

# Microbial metabolism: optimal control of uptake versus synthesis

Steven A. Frank

Department of Ecology and Evolutionary Biology, University of California, Irvine, CA, USA

## ABSTRACT

Microbes require several complex organic molecules for growth. A species may obtain a required factor by taking up molecules released by other species or by synthesizing the molecule. The patterns of uptake and synthesis set a flow of resources through the multiple species that create a microbial community. This article analyzes a simple mathematical model of the tradeoff between uptake and synthesis. Key factors include the influx rate from external sources relative to the outflux rate, the rate of internal decay within cells, and the cost of synthesis. Aspects of demography also matter, such as cellular birth and death rates, the expected time course of a local resource flow, and the associated lifespan of the local population. Spatial patterns of genetic variability and differentiation between populations may also strongly influence the evolution of metabolic regulatory controls of individual species and thus the structuring of microbial communities. The widespread use of optimality approaches in recent work on microbial metabolism has ignored demography and genetic structure.

## INTRODUCTION

Each microbial species takes up particular compounds and releases others. Biochemical fluxes between species determine resource flows through the community. To understand how each species reacts to and in turn influences other species, one must find key attributes of biochemical fluxes that bring the diffuse interconnected complexity into sharp focus.

One focal point arises from the complex organic molecules required by many organisms. For example, several microbes require vitamin $B_{12}$ (*Roth et al., 1996*). Only certain species make that vitamin via an intricate biosynthetic pathway that requires cobalt, often a rare and potentially limiting factor. Among those species that require $B_{12}$, some cannot make it and must take it up, some can make it but cannot take it up, and some can switch between uptake and synthesis (*Bertrand et al., 2012*).

Varying patterns of uptake and synthesis occur for different molecules. Each species evolves its characteristics in response to external availability and internal need. Interactions between species cause evolutionary and ecological feedbacks that shape patterns of resource flow through the community.

In this article, I analyze a simple mathematical model for the tradeoff between uptake and synthesis of a molecule that limits growth. I study the evolutionary response of a

Corresponding author
Steven A. Frank, safrank@uci.edu

single species to external availability and internal decay. In additional, the overall population varies demographically with regard to how long local patches last before extinction. I particularly emphasize the demographic aspect, because prior work on the evolution of metabolic regulation rarely accounts for the key ways in which spatial and temporal variations in resources, survival and reproduction shape evolutionary response (*Frank, 2010d*).

The model shows that, under many conditions, species switch sharply between uptake with no internal synthesis and internal synthesis with no uptake. Pure uptake means dependence on production by other species, partitioning the community into producers and nonproducers. Pure synthesis means that the focal species may become a source for other species. Some conditions lead to a mixture of uptake and synthesis. In that case, individuals maintain costly internal production but also scavenge the externally available molecules released by dying cells.

The model illustrates the kinds of scaling relations that influence each species and thus contribute to the structuring of communities. For example, influx rates from external sources matter only in relation to rates of outflux, internal decay, and the cost of uptake. Aspects of demography also matter, such as cellular birth and death rates, the expected time course of a local resource flow, and the associated lifespan of the local population. I discuss how spatial patterns of genetic variability and differentiation between populations may strongly influence the evolution of metabolic regulatory controls of individual species and thus the structuring of microbial communities.

My main point is that demography and the genetic structure of populations must be very important in shaping the metabolic properties of species and communities (*Frank, 1996*; *Crespi, 2001*; *Pfeiffer, Schuster & Bonhoeffer, 2001*; *West et al., 2007*; *Frank, 2010a*; *Frank, 2010b*; *Frank, 2010d*; *Frank, 2010c*; *Frank, 2013*). However, the widespread use of optimality approaches in recent work on microbial metabolism has almost universally ignored demography and genetic structure (*Ebenhoh & Heinrich, 2001*; *Schuetz, Kuepfer & Sauer, 2007*; *Banga, 2008*). Instead, that recent work has mostly used either growth rate or biomass yield as the objectives optimized by natural selection. Even the advanced multi-objective optimizations of the most sophisticated recent analyses ignore demography and genetic variability (*Handl, Kell & Knowles, 2007*; *Sendin et al., 2010*; *Higuera et al., 2012*; *Schuetz et al., 2012*).

## METHODS

The model analyzes an isolated species' metabolic design. Fitness optimization provides specific predictions about the tradeoff between uptake and synthesis. The analysis uses discounted population size as the measure of fitness (*Fisher, 1930*; *Stearns, 1992*; *Frank, 2010d*). The idea is that, at any time, the long-term contribution of a genetic clone to the future of the population depends on the number of cells in that clone.

The total fitness of a clone over its full life cycle in a resource patch is the size of the clone at each time multiplied by the probability that the clone survives to that time. Later times in the life cycle are discounted because the probability of survival to a particular

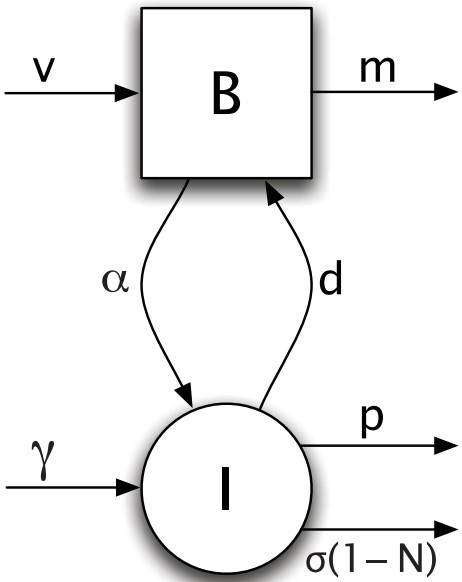

**Figure 1** **Flows of a metabolic factor between internal cellular stores, *I*, and the external environmental store, *B*.** See Table 1 for descriptions of parameters. Based on system dynamics in Eq. (1).

time is a decreasing function of the amount of time that has passed. Thus, factors that influence the survival of clones and the overall demography of the population strongly affect fitness and how evolutionary process shapes particular metabolic tradeoffs.

In the Discussion, I consider an extended measure of fitness that accounts for genetic variability between competing cells in a local population. Genetic variability often significantly alters the objective fitness measure and associated optimal traits (*Hamilton, 1970*; *Frank, 1998*), and therefore has major consequences for metabolic design (*Pfeiffer, Schuster & Bonhoeffer, 2001*; *Frank, 2010d*).

I focus on a particular organic compound that affects cellular growth rate. Various fluxes determine the flow of the compound through the local population. Fig. 1 shows the division of fluxes into distinct compartments, which include the extracellular environment, *B*, in the local population (patch), the internal cellular environment, *I*, of the cells in the patch, and compartments external to the patch.

Independently of the focal cells in a patch, influx of the compound from external sources, *v*, is balanced by outflux, *m* (Fig. 1). In the absence of local cells, the external concentration comes to an influx-outflux equilibrium. The compound flows into cells via cellular uptake, $\alpha$, and is released from cells when they die, at rate *d*. Cells can make the compound by de novo internal synthesis, at rate $\gamma$. Within cells, the compound decays at rate *p*. The internal concentration per cell is reduced following cellular division, because the existing molecules must be split between the daughter cells. The rate of dilution by cell division is $\sigma(1 - N)$.

**Table 1** Variables and parameters, see Appendix for nondimensional scalings.

| | |
|---|---|
| State variables: | |
| $N$ | number of cells in local population |
| $B$ | number of molecules of metabolic factor outside of cells |
| $I$ | number of molecules of metabolic factor within each cell |
| $t$ | nondimensional time scale |
| Control variables: | |
| $\alpha$ | external uptake rate of metabolic factor |
| $\gamma$ | internal synthesis rate of metabolic factor |
| Parameters: | |
| $a$ | cost for uptake via diminished population growth rate |
| $g$ | cost for synthesis via diminished population growth rate |
| $d$ | intrinsic cellular death rate |
| $p$ | loss rate of internal molecules of metabolic factor |
| $v$ | extrinsic inflow of metabolic factor |
| $m$ | loss rate of external molecules of metabolic factor |
| $u$ | patch death rate, with average patch survival $1/u$ |
| Other processes: (see Appendix) | |
| $c$ | scaling for molecules of metabolic factor released at death |
| $k$ | scaling for molecules of metabolic factor taken up by cells |

The next section provides the equations that govern the dynamics of the compound fluxes and cellular growth. The following section finds the optimal tradeoff between uptake and synthesis for a genetically uniform clone under different assumptions about flux rates and demography set by patch survival rates. The Discussion considers how genetic variability within patches affects the optimal tradeoff between uptake and synthesis.

## RESULTS

### Dynamics

I focus on the evolution of two control variables: the extracellular uptake of a metabolic factor at rate $\alpha$, and the intracellular synthesis of that metabolic factor at rate $\gamma$. Three variables define the state of the system: the number of cells in the local population, $N$; the number of molecules of the metabolic factor outside of the cells, $B$; and the number of molecules of the metabolic factor within each cell, $I$. The dynamics also depend on several parameters listed in Table 1.

To simplify the analysis, I scale all variables and parameters into nondimensional form. For example, I express population size, $N$, as a fraction of the maximum population size that can be attained, and I express the number of internal molecules, $I$, as a fraction of the amount required to achieve one-half of maximum growth rate. The Appendix shows the full expression of dynamics in terms of all of the dimensional values and the translation into the scaled nondimensional forms given in the main text. In the following, when I describe the number of molecules or the rate of a process or the change in time, those values are understood to be nondimensionally scaled relative to some baseline number or rate.
Table 1 shows all of the nondimensional variables and parameters. Using those terms, the scaled nondimensional dynamics are

$$\dot{N} = [\sigma(1 - N) - d]N \tag{1a}$$

$$\dot{B} = v + dIN - \alpha BN - mB \tag{1b}$$

$$\dot{I} = (\alpha B + \gamma) - pI - \sigma(1 - N)I, \tag{1c}$$

with maximum birth rate, $\sigma$, of

$$\sigma = \left(\frac{I}{1 + I}\right) - (a\alpha + g\gamma).$$

Fig. 1 shows the flows of the metabolic factor between the internal store within cells, $I$, and the external store, $B$.

## Clonal structure with variable demography

Suppose that spatially distinct resource patches come and go. A patch could be a host, a decaying organism, or a sugary runoff. In this section, I assume that a patch may become colonized by a small genetically uniform clone. The clone grows and sends out dispersers to colonize new patches. Eventually the patch dies off.

The fitness of a clone in a particular patch is the total number of dispersers sent out of the patch. To calculate fitness, I assume that, at any time, the rate of successful dispersers out of a patch is proportional to the population size in the patch. A patch has a constant death rate, $u$, and average survival time of $1/u$. Thus, total expected fitness over a patch life cycle is

$$w = \int_{t=0}^{\infty} N(t)e^{-ut}\,dt, \tag{2}$$

which is a classic expression for fitness from life history theory (*Fisher, 1930*; *Stearns, 1992*), and was used by Frank (*Frank, 2010d*) to study microbial metabolism. I calculated the optimal control rates for uptake, $\alpha$, and synthesis, $\gamma$. Using the dynamics in Eq. (1), I optimized fitness in Eq. (2) by the computational method of differential evolution (*Storn & Price, 1997*).

Seven parameters (Table 1) influence the optimal control values presented in Fig. 2. The axes of each plot show combinations for the costs of uptake and synthesis, $a$ and $g$. The different plots vary the values of the inflow and outflow parameters, $v$ and $m$. The list at the top of the figure shows the values for the patch lifespan, $1/u$, cell death rate, $d$, and decay rate within cells, $p$.

Higher total costs, $a + g$, do not strongly affect the ratio of uptake versus synthesis, $\log(\alpha/\gamma)$. Both control variables tend to decline with a rise in costs. In some cases, synthesis declines more rapidly than uptake, causing a rise in $\log(\alpha/\gamma)$.

A decrease in the ratio of costs, $\log(a/g)$, typically favors a rise in the ratio of uptake versus synthesis, $\log(\alpha/\gamma)$. Exceptions sometimes occur for high total costs. For example, in the lower left panel of Fig. 2, when total costs $a + g$ are high, a decline in $\log(a/g)$ first

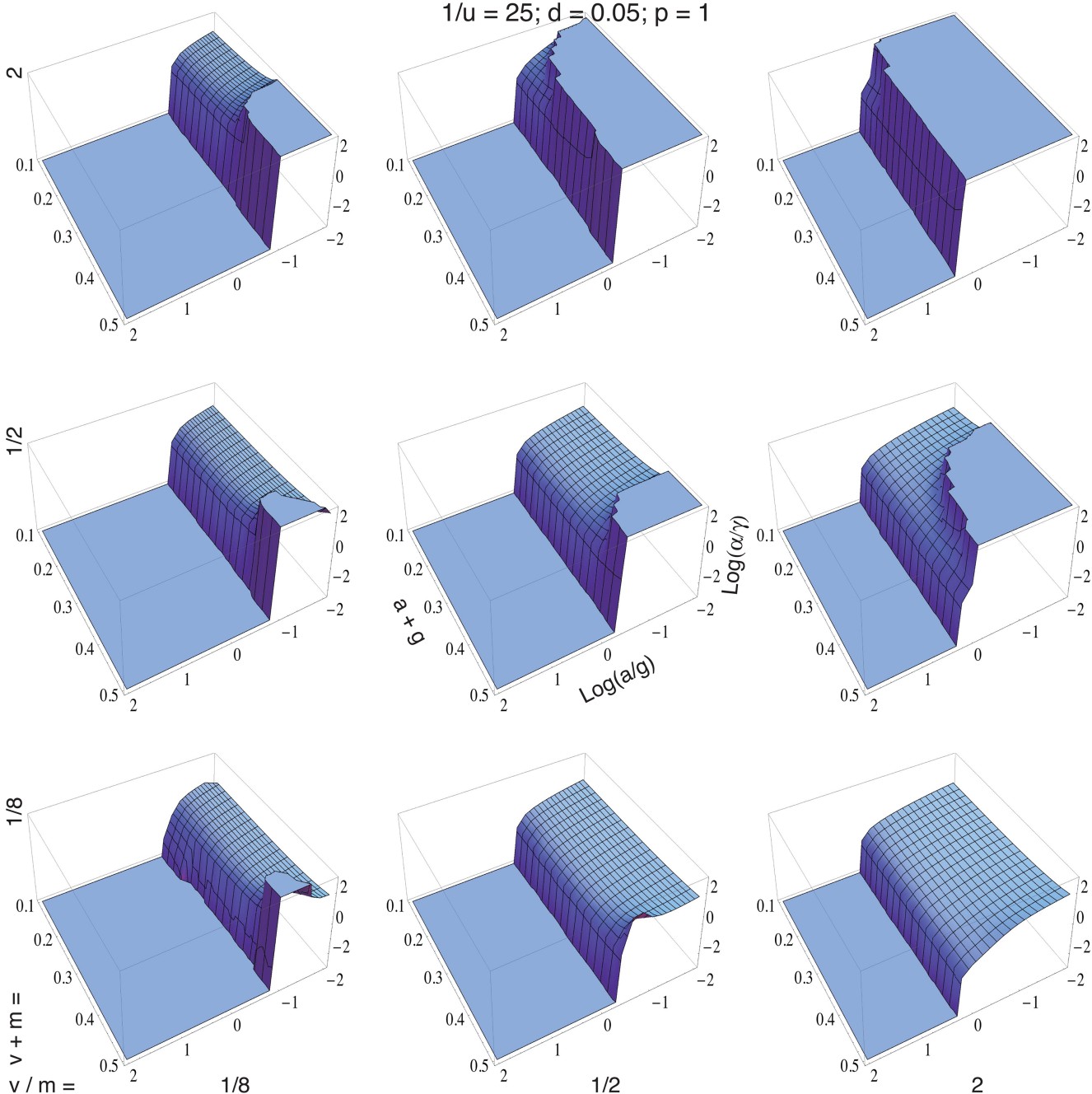

**Figure 2 Optimal control variables for uptake, $\alpha$, and synthesis, $\gamma$.** Optimal values maximize fitness in Eq. (2) using the dynamics from Eq. (1). Each patch is purely clonal. The center panel shows the labeling for axes. All logarithms use base ten. The height of each surface, $\log(\alpha/\gamma)$, scaled between $-3$ and $3$, shows the relative magnitude of uptake versus synthesis. The flat regions show values outside of that range. For the costs of uptake and synthesis, $a$ and $g$, it is convenient to present the two dimensions as the sum and the ratio of the parameters. The same dimensional split into sum and ratio also applies to the inflow and outflow parameters, $v$ and $m$. Initial values for the state variables are $N = 10^{-5}$, $B = v/m$, and $I = (\alpha B + \gamma)/p$.

causes a sharp rise in the ratio of uptake versus synthesis and then a decline in that ratio. Associated with the sharp rise, the synthesis rate declines to nearly zero under conditions that favor uptake and little synthesis.

As the cost ratio continues to decline so that $a$ is dropping rapidly and $g$ is rising slowly, the uptake rate continues to rise as expected. Interestingly, as the uptake rate continues to rise, internal synthesis also starts to increase. Under those conditions, uptake is sufficiently high that the clone gains from costly internal synthesis, because, with sufficiently high uptake rate, one cell can recycle the internally produced metabolic factor that is released by death of another cell, leading to synergism between uptake and synthesis.

An increase in the sum of influx and outflux, $v + m$, occurs as one moves up a column of plots in Fig. 2. There tends to be a sharper switch between uptake and synthesis with high $v + m$ and greater dominance by external flows of the key metabolic factor. At high $v + m$, either there is enough of the factor available externally or there is not. Recycling of the internal stores from cell death has little effect, because the external concentration is dominated by extrinsic flows.

An increase in the ratio of external influx to outflux, $v/m$, occurs as one moves to the right across a column of plots in Fig. 2. Relatively stronger influx favors greater uptake by providing more of the factor available externally.

Fig. S1 shows the consequences of varying the average patch lifespan, $1/u$, and the cell death rate, $d$. Longer lasting patches favor relatively more uptake at higher cost ratios of uptake versus synthesis. Longer lasting patches also favor relatively more synthesis when the relative cost of synthesis rises and the cost ratio of uptake versus synthesis declines. It may be that longer patch lifespan provides more opportunity for synergism between uptake and synthesis, slowing the change in relative dominance by uptake versus synthesis.

The plot arrays on the right side of Fig. S1 show a higher value of the cellular death rate, $d$. Greater cell death interacts with several other parameters to influence the ratio of uptake versus synthesis.

Fig. S2 shows the consequences of increasing $p$, the rate of decay of the metabolic factor within cells. High $p$ reduces the potential for synergism between uptake and synthesis. When the internal decay is fast, then the clone cannot gain much from uptake of the factor released from dying cells, because the internal decay within cells is too high and so less is released upon death. For higher values of $p$, lack of synergism between uptake and synthesis leads to a sharper switch between dominance by uptake versus synthesis.

Mixed uptake and synthesis occur for some parameter combinations. Mixed expression seems to depend primarily on synergistic interactions between uptake and synthesis, in which internal production by a cell followed by cell death aids neighboring cells that take up the released products. Thus, synergism seems to be favored when internal production, a trait that is costly to the individual, provides a benefit to a genetically related neighbor. If so, then greater genetic variability within patches should reduce the fitness benefit of aiding neighbors and thus reduce the synergistic effect (see Discussion).

Other conditions may also favor mixed expression of uptake and synthesis. For example, averaging over variable environments that sometimes favor uptake and sometimes favor synthesis may lead to mixed expression, particularly when the cells cannot switch with sufficient precision between uptake and synthesis in response to changes in external conditions.

## DISCUSSION

Many recent studies of microbial metabolism use optimization methods (*Schuetz, Kuepfer & Sauer, 2007*). Those studies consider how different aspects of regulatory control influence success. The idea is that natural selection tends to favor maximum success subject to constraints that limit possible combinations of traits. Optimality methods provide a way to interpret data on regulatory control with respect to how particular traits contribute to success or how those traits are limited by specific constraints.

Analysis of optimality requires choice of a particular measure of success. The measure of success may have multiple dimensions, with tradeoffs between dimensions. For example, a simple multi-objective function for success considers the tradeoff between growth rate and biomass yield (*Pfeiffer, Schuster & Bonhoeffer, 2001*).

In this article, I analyzed the tradeoff between uptake and synthesis of complex organic compounds. I showed how optimization of success may lead to different combinations of uptake and synthesis, most often with a sharp switch between relative dominance by uptake or synthesis. The balance between alternatives depends on several conditions, such as the inflow and outflow of the compound from the extracellular environment or the rate of cellular death.

The measure of success for optimization has a very strong effect on the predictions. I used a measure that considers total reproduction (yield) discounted by time. My measure sums each time point from the founding of a local colony to the eventual death of that colony. At each time, the success is the number of cells in the colony discounted by the probability that the colony survives to that time. Thus, earlier reproduction is weighted more heavily than later reproduction, providing a measure that weights rate versus yield according to the time discount parameter.

Key metabolic tradeoffs, such as rate versus yield or uptake versus synthesis, inevitably depend on the discounting of future reproduction. In this study, strong discounting, associated with short average colony lifetimes, typically favored a sharper transition between relative dominance by uptake or synthesis (Fig. S2).

Most optimality studies of microbes use either growth rate or biomass yield as the objective function to be optimized, ignoring the demographic consequences of time discounting (*Schuetz et al., 2012*). Time discounting has strong consequences and is likely to be nearly universal (*Frank, 2010d*). Thus, the many studies that ignore such demographic factors must be missing an essential force in the evolutionary design of microbial regulatory control.

Recent optimality studies of microbes typically optimize clonal success (*Schuetz et al., 2012*). However, analyses limited to clonal success may be misleading. Several studies

have shown the very strong and inevitable ways in which patterns of genetic variability affect the evolutionary design of microbial regulatory and growth related traits (*Frank, 1996*; *Pfeiffer, Schuster & Bonhoeffer, 2001*; *Gardner, West & Griffin, 2007*; *West et al., 2007*; *Frank, 2010c*; *Frank, 2010d*).

In a prior study, I showed a simplified way to approximate the role of genetic variability in optimality analyses of microbes (*Frank, 2010d*) (see Supplemental Information). In that method, one starts with a particular common genotype and an alternative rare genotype. When genetic mixtures occur, the common genotype is almost always with another common genotype. Thus, one can calculate the aggregate fitness of the common type by analyzing its success as a clone.

By contrast, the rare type will occur in two different kinds of patches. With probability $r$, the rare type will settle in a patch with another rare type, and with probability $1 - r$, the rare type pairs with the common type. The aggregate fitness of the rare type is the average of the two patch compositions. Here, $r$ is the spatial correlation between genotypes within patches, which is equivalent to the genetic coefficient of relatedness used in studies of kin selection and social evolution (*Hamilton, 1970*; *Frank, 1998*). More mixing between genotypes reduces $r$.

A possible optimal type would be one for which fitness when common is greater than any rare alternative (*Maynard Smith, 1982*). Here, optimality simply means evolutionary stable when common against any rare genetic variant. Although this idea is simple and works well for many problems, it can be technically challenging to find optima for certain problems. For example, the joint optimization of uptake and synthesis requires simultaneous optimization in two dimensions. Joint optimization is in principle easy to do, but challenging in practice because of numerical complexities. Here, I limit my discussion to a few conjectures about how genetic variability might influence uptake versus synthesis.

In the Results, I presented several examples in which synergism between uptake and synthesis seemed to influence the optimal trait values. In those examples, internal synthesis apparently provides an extra advantage to a cell because, when the cell dies, it releases its stored internal metabolic factor, which then may be taken up by genetically identical neighboring cells. Presumably that advantage would decline as genetic mixing lowered $r$, the genetic correlation between neighboring cells. Synergism appeared to be a powerful factor under certain circumstances. Thus, a significant interaction may arise between the genetic structure of populations and the synthesis-release-uptake cycle.

In general, interactions often arise between genetic structure and traits that influence competition or cooperation between cells (*Frank, 1996*; *Crespi, 2001*; *Pfeiffer, Schuster & Bonhoeffer, 2001*; *West et al., 2007*; *Frank, 2010a*; *Frank, 2010b*; *Frank, 2010c*; *Frank, 2010d*; *Frank, 2013*; *Diard et al., 2013*). Such interactions can be analyzed by optimality and other analytical methods only when genetic structure is included explicitly as an aspect of the target objective function, which defines fitness. The results presented in this article and earlier studies also showed that time discounting also can strongly influence fitness in the context of microbial tradeoffs in the regulatory control of metabolism. Time

discounting is a particular kind of demographic process. Evolutionary analyses of life history generally show strong effects of many demographic processes.

Interaction between demography and genetic structure are common and potentially very strong (*Frank, 1998*; *Frank, 2010a*; *Frank, 2010d*). Further progress in using optimality to study microbial regulation and metabolism will require wider use of demographically and genetically realistic objective functions.

## APPENDIX. NONDIMENSIONAL DEFINITIONS

The dynamics of Eqs. (1a)–(1c) expressed in terms of dimensional variables and parameters are

$$\dot{N} = [\sigma(1 - N/K) - d]N \tag{3a}$$
$$\dot{B} = v + cdNI - kN\alpha B - mB \tag{3b}$$
$$\dot{I} = (\alpha B + \gamma) - pI - \sigma(1 - N/K)I \tag{3c}$$

with maximum birth rate

$$\sigma = b\left(\frac{I}{s+I}\right) - (a\alpha + g\gamma/s).$$

The actual birth rate is the maximum, $\sigma$, devalued by $1 - N/K$. The discount arises by the competition among cells over resources not explicitly included in the model.

All variables and parameters here are dimensional, and the time scaling is with respect to the dimensional measure of time, $\tau$. The following substitutions transform the nondimensional system in Eqs. (1a)–(1c) to the dimensional system in Eqs. (3a)–(3c).

The nondimensional timescale is $t = \tau b$. For each of the following substitutions, the left side is a nondimensional expression and the right side is a dimensional expression: $N = N/K$; $B = B/s$; $I = I/s$; $\alpha = \alpha/b$; $\gamma = \gamma/sb$; $d = d/b$; $v = v/sb$; $c = cK$; $k = kK$; $m = m/b$; and $p = p/b$. The terms $a$ and $g$ are nondimensional cost scalings. We can express the nondimensional value of $\sigma$ in terms of the nondimensional definitions for the other terms as

$$\sigma = \left(\frac{I}{1+I}\right) - (a\alpha + g\gamma).$$

The parameter $c$ is a scaling factor for the number of molecules released when a cell dies, and the parameter $k$ is a scaling factor for the number of molecules removed from the external source when taken up by cells. Those scaling factors can be helpful in analyzing the details of particular systems. In this article, I emphasize the general structure of the problem rather than the quantitative details of particular systems. Therefore, I have set $c = k = 1$ in the main text, dropping those parameters from the analysis.

The patch death rate is $u$. The text uses the nondimensionally scaled expression $u = u/b$, where the left-hand side is by convention the nondimensional version of the dimensional scale on the right-hand side.

### Funding

National Science Foundation grants EF-0822399 and DEB-1251035 support my research. The funders had no role in study design, data collection and analysis, decision to publish, or preparation of the manuscript.

### Grant Disclosures

The following grant information was disclosed by the authors:
National Science Foundation grants EF-0822399 and DEB-1251035.

### Competing Interests

The author declares that he has no competing interests.

### Author Contributions

- Steven A. Frank conceived and designed the experiments, performed the experiments, analyzed the data, contributed reagents/materials/analysis tools, wrote the paper, prepared figures and/or tables, reviewed drafts of the paper.

### Supplemental Information

Supplemental information for this article can be found online at
http://dx.doi.org/10.7717/peerj.267.

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
