# Peer review of "Microbial metabolism: optimal control of uptake versus synthesis"

_PeerJ, doi:10.7717/peerj.267_

## Round 0.1 · original submission · Minor Revisions

Two referrees reviewed the article and both found that this is great piece of work. Referee 1 had a few suggestions as to further improve the manuscript that should be easily taken into account in a revised version.

Stuart West ·

Basic reporting

See below

Experimental design

See below

Validity of the findings

See below

Additional comments

This paper has an excellent focus, examining the trade off between uptake and synthesis in microbes. It develops theory that clarifies conceptual issues and will guide future empirical work, with comparative predictions clearly laid out. The case for the importance of demography is clear and well made.

1. The assumption of clonal structure is strong, but well justified for this starting step. The potential implications of genetic diversity are mentioned, but i was left wondering how it might influence the extent to which you get a sharp step between uptake and synthesis (versus mixed expression). Just more likely?

2. The seven parameters (Table 1) are considered to be independent (open model). How justified is this? Are there some we might expect to be correlated across/within species?

3. Fig 2 is key. Both labels and legend could use words as well as parameter labels.

4. Supp fig 1 & 2. Why can’t some of the key results be in actual paper?

5. Variable environment seems key, especially whether it will favour mixed expression. Some results or more discussion?

6. Conditional expression. This seems highly likely (as with exofactors). Why not expand and put in main text?

·

Basic reporting

Meets PeerJ standards.

Experimental design

Meets PeerJ standards.

Validity of the findings

Meets PeerJ standards.

Additional comments

Optimization thinking and approaches are typically employed in the theoretical understanding of microbial metabolism. This paper emphasizes the neglected role of demography in mediating this optimization. A simple model is developed and analysed, revealing how varying demographic assumptions can have a dramatic impact on how microbial strains are favoured to take up, synthesize and/or release nutrients. The paper is beautifully written and clear, and I can see it opening up a very productive avenue of research activity.

As is my policy, I waive anonymity

Andy Gardner

---

## Round 0.2 · accepted · Accept

Nothing to add. Thank you for this nice piece of work.